# Controlled Molecular Arrangement of Cinnamic Acid in Layered Double Hydroxide through pi-pi Interaction for Controlled Release

**DOI:** 10.3390/ijms25084506

**Published:** 2024-04-19

**Authors:** Taeho Kim, Seung-Min Paek, Kang-Kyun Wang, Jin Kuen Park, Fabrice Salles, Jae-Min Oh

**Affiliations:** 1Department of Energy and Materials Engineering, Dongguk University, Seoul 04620, Republic of Korea; taeho0408@naver.com; 2Department of Chemistry, Kyungpook National University, Daegu 41566, Republic of Korea; 3BI Bio-Photonics Co., Ltd., Gimpo-si 10090, Republic of Korea; biprimelab2020@gmail.com; 4Department of Chemistry, Hankuk University of Foreign Studies, Yongin 17035, Republic of Korea; jinkpark@hufs.ac.kr; 5Institute Charles Gerhardt des Matériaux (ICGM), Université de Montpellier, CNRS, ENSCM, 34090 Montpellier, France

**Keywords:** layered double hydroxide, molecular arrangement, phenolic acid, cinnamic acid, π-π interaction, release, intermolecular interaction

## Abstract

Cinnamic acid (CA) was successfully incorporated into Zn-Al layered double hydroxide (LDH) through coprecipitation. The CA moiety was stabilized in the interlayer space through not only electrostatic interaction but also intermolecular π-π interaction. It was noteworthy that the CA arrangement was fairly independent of the charge density of LDH, showing the important role of the layer–CA and CA-CA interactions in molecular stabilization. Computer simulations using the Monte Carlo method as well as analytical approaches including infrared, UV-vis spectroscopy, and differential scanning calorimetry showed the existence of intermolecular interaction. In order to reinforce molecular stabilization, a neutral derivative of CA, cinnamaldehyde (CAD), was additionally incorporated into LDH. It was clearly shown that CAD played a role as a π-π interaction mediator to enhance the stabilization of CA. The time-dependent release of CA from LDH was first governed by the layer charge density of LDH; however, the existence of CAD provided additional stabilization to the CA arrangement to slow down the release kinetics.

## 1. Introduction

Recent advances in nanotechnology have enabled elaborate control of molecular or particle arrangements on a certain substrate or in a free-standing form. The artificial arrangement of molecules or particles enables the versatile functionalities of materials. For example, manipulating molecular arrangements realizes various structure colors; the nematic cellulose nanocrystals in a film were reported to alter the arrangement of cellulose depending on the amount of water molecules, exhibiting a colorimetric response against humidity [1]. The particle arrangement of silver nanoparticles on a metal hydroxide substrate could be utilized in the surface-enhanced Raman scattering sensor to detect trace amounts of organic materials [2]. The molecular arrangement in a two-dimensional direction has unique advantages in terms of applications. For instance, two-dimensional assemblies of molecular switches and rotors provided better manipulation of molecular machines than isolated free molecules [3]. Controlling the arrangement of naphthalene derivatives along a two-dimensional direction exhibited asymmetric light propagation, which could be further applied to optical plane diodes [4].

Among the various techniques to develop planar arrays of molecules, the most easily accessed method is to utilize a two-dimensional substrate. In this regard, layered materials with expandable interlayer spaces, such as layered double hydroxide (LDH), have been utilized. Especially, LDH has been highlighted in biological applications such as drug delivery or cosmetics due to its excellent ability to accommodate bio-functional molecules in its interlayer space in a controlled manner. LDH consists of positively charged nanolayers with the chemical formula [M(II)_1−x_M(III)_x_(OH)_2_]^x+^ and negatively charged interlayer anions [5]. Due to the periodic positive charge and exchangeability of interlayer anions, LDH has been widely applied to immobilize biologically functionalized molecules and to release payloads in a controlled manner [6,7,8]. Compared with well-known polymer-based drug delivery systems [9], LDH has distinguishable advantages and disadvantages. Polymer-based drug delivery materials can be developed in a wide variety by modifying the side chains or arranging monomers. It is easy to control the hydrophilicity/hydrophobicity of the material, and the release properties of polymers are fairly ordered according to the size of the particles. However, the degradation of polymer-based delivery systems is not controllable, giving rise to unexpected residues and byproducts. On the other hand, LDH-based delivery systems do not have the flexibility of polymers, and it is not simple to prepare them with the intended physicochemical properties. Nevertheless, once prepared, LDH delivery systems can efficiently prevent payloads from external chemical/physical stimulation; the degradation of LDH results in bioresorbable cations and anions, giving rise to less toxicity. In this regard, LDH-based delivery systems are also attracting interest, along with polymer-based systems.

The molecular immobilization and release properties of organic–LDH hybrids have been studied in terms of controlling physicochemical parameters, e.g., chemical composition, as well as applying different synthetic routes. For instance, vanillic acid was incorporated into ZnAl-LDH with different metal ratios [10]. As the charge density of ZnAl-LDH was influenced by the Zn/Al ratio, the release of incorporated molecules could be different depending on the metal composition; a higher charge density facilitated the faster release of vanillic acid in an aqueous medium. The synthetic route of the organic–LDH hybrid also influenced the molecular arrangement in the interlayer space and release pattern. Kang et al. incorporated ferulic acid into LDH through three different methods, namely, ion exchange, reconstruction, and exfoliation–restacking [7]. Although all the organic–LDH hybrids had the same metal composition and particle dimensions, the synthetic condition subtly controlled the molecular conformation, giving rise to different release kinetics. As the release of payloads from LDH was not entirely controlled by the physicochemical properties alone, additional surface coating or composite strategies were also utilized. The enteric coating of a drug–LDH hybrid with an acrylate polymer, Eudragit^®^ S100 or L100, is a representative example of how to block drug release in acidic conditions and facilitate release under neutral conditions [11]. The preparation of a composite of a drug–LDH hybrid and a hydrogel like alginate is also applicable to control the release kinetics of drug molecules depending on pH conditions [12].

Although there have been various approaches to control molecular arrangement in LDH to consequently regulate the release pattern, it has not been easy to prepare the molecular arrangement as intended. As the strong electrostatic interaction between organic molecules and LDH layers governs most of the molecular arrangement, the relatively weak intermolecular interaction could not play a role in the molecular array. In this study, we are going to suggest a simple way to control the molecular arrangement in LDH’s gallery space by utilizing a neutral derivative of the anionic organic molecule. As the neutral molecule does not interact with the LDH layer, it can play an important role in the intermolecular interactions.

Considering the wide utilization of phenolic acid as a biofunctionalized substance, cinnamic acid (CA) was selected as a model organic molecule. The stabilization of CA in an inorganic matrix could be applied to cosmetics for absorbing ultraviolet rays and to nutraceuticals for efficiently preserving the CA moiety during food processing. In order to induce a molecular arrangement in LDH, three kinds of ZnAl-LDHs with different Zn/Al ratios were prepared; the layer charge density of LDH is highly governed by the Zn/Al ratio. After the CA-LDH (CL) hybrids were prepared, cinnamaldehyde (CAD), a neutral CA derivative, was introduced into the gallery space in order to add an additional molecular interaction. We examined the structure and molecular arrays of CA and CAD in CA-LDH-CAD (namely, CLC in this research) through experimental approaches (X-ray diffraction, infrared spectra, UV-vis, differential scanning calorimetry, etc.) as well as a theoretical approach utilizing computational simulation. Taking into account the potential applications in cosmetics and nutraceuticals, the time-dependent release profile of CA in deionized water was monitored, and the importance of the intermolecular interaction between CA and CAD inside the LDH matrix was analyzed.

## 2. Results and Discussion

### 2.1. X-ray Diffraction Patterns and Computational Simulations

The lattice expansion along the c-axis observed in the X-ray diffraction patterns of the CL and CLC hybrids implied that both CA and CAD were successfully incorporated in the interlayer space of [Zn_1−x_Al_x_(OH)_2_]^x+^ LDH. In addition, both the CL and CLC hybrids showed characteristic lattice peaks of LDHs such as (012) and (110) at 33.7° and 59.7°, respectively, suggesting that the LDH structure was not altered through successive incorporation of CA and CAD (Figure 1). In order to calculate the d-spacing along the crystallographic c-axis, Bragg’s equation, which is defined as nλ = 2dsinθ (where d is the d-spacing corresponding to a specific lattice plane; n is an integer (1 was applied in this study); lambda is the wavelength of X-ray (0.154 nm for Cu Kα in this study); and θ is the Bragg angle), was applied for the (003) peak, as the layer stacking of LDH is rhombohedral—three layers make one unit cell. Comparing with the (003) peak of pristine ZnAl-NO_3_-LDH corresponding to a d-spacing of 0.08 nm, the (003) peaks of all the CLs shifted to the lower 2θ region. The d-spacing values of the CLs were determined to be 1.85 nm, which corresponded to the inter-digitated double-layer arrangement of CA molecules in LDH’s gallery space, taking advantage of π-π interaction through parallel stacking (Figure 2).

In spite of their comparable molecular dimensions, the reported d-spacing values of phenolic acid (vanillic acid, cinnamic acid, gallic acid, ferulic acid, etc.)-intercalated LDH were fairly diverse, implying their uncontrollable manner of interlayer arrangement. It was reported that the d-spacing of gallic acid-intercalated LDH was ~1.1 nm, suggesting a lying-down arrangement [13]. Vanillic acid-intercalated LDH was reported to have ~1.5 nm of d-spacing [10,13], implying a single-layer arrangement in the gallery space. Ferulic acid exhibited d-spacing expansion to ~1.7 nm, which corresponded to the inter-digitated double-layer arrangement [7,14]. Meanwhile, cinnamic acid-intercalated LDH showed both 1.4 nm [15] and 1.8 nm [16] of d-spacing, which corresponded to a tilted single-layer and an inter-digitated arrangement, respectively. It seems that the molecular arrangement is mainly dependent on layer–molecule interactions and is minorly affected by intermolecular interactions. The current CL would be arranged in an inter-digitated manner in which the π-conjugation of CA effectively interacts with the adjacent molecule, taking advantage of intermolecular interactions.

In order to support the interlayer arrangement of CA, we carried out Monte Carlo simulations at ambient temperature, as described in the computational section. As shown in Figure 3a, the CA moiety interacts with the LDH layer through electrostatic interaction at a distance of 0.21~0.23 nm. Taking into account the molecular dimension of CA, 0.8 nm from the benzene end to the carboxylate, and the layer thickness of 0.48 nm of LDH, the CA moiety could be arranged as displayed in Figure 2a. It is worth noting that the d-spacing of CL was similar across different samples, regardless of the Zn/Al ratio. The theoretical demanded area of positive charge in LDH can be calculated based on the unit cell diagram along the ab-plane, as shown in Appendix A. The LDH with a Zn/Al ratio of 2/1 produces one positive charge in a 0.25 nm^2^ hexagon (CL1). The demanded area increases by 5/3 and 8/3, respectively, for Zn/Al ratios of 4/1 and 7/1, resulting in demanded area values of 0.42 (CL2) and 0.67 nm^2^ (CL3). The cross-sectional area of the CA molecule was approximately 0.23 nm^2^; in other words, the LDH with a ratio of 2/1 provides the exact space for CA molecules, while the other LDHs give more space to CA molecules than required. Usually, when the area of the layer is larger than the molecular dimensions, it often results in the tilting of the interlayer molecules; however, the strong π-π stabilization in CL could keep the upright arrangement of the CA moiety. This result is supported by Monte Carlo simulations, as observed in Figure 3b.

Upon additional incorporation of CAD, the lattice of CL expanded again along the crystallographic c-axis, showing a final d-spacing of 2.46 nm (Figure 1). The incorporation of CAD is thought to open the interlayer space, by which way the CAD moieties could be stabilized between the CA arrangements, taking advantage of the π-π interaction between CAD and CA (Figure 2). This phenomenon was also supported by the Monte Carlo simulation, in which CAD-CAD and CA-CAD interactions were observed (see orange and purple values in the snapshots in Figure 3c). As in the CA-CA interaction, the π-conjugation in CL and CAD can stabilize each other by π-π stacking with 0.35–0.37 nm distances. It was noteworthy that the interlayer distance of CLC was also similar across samples, regardless of the Zn/Al ratio. This implies that the required amount of CAD was incorporated into the interlayer space of CL to hold the CA molecules attached to the upper and lower layers of LDH. Through X-ray diffraction patterns, we could confirm that both the CA and CAD moieties were successfully stabilized between LDH layers through electrostatic interaction (between CA and LDH) and π-π interaction (between CA and CAD).

### 2.2. Fourier Transform Infrared Spectroscopy

The intact structures of CA or CAD in the hybrids were confirmed by Fourier transform infrared (FT-IR) spectra (Figure 4). The spectrum of anionic CA (Figure 4a), which was obtained by deprotonation of CA, showed characteristic peaks at ~1390 cm^−1^ (νs(COO-) in carboxylate), 1550 cm^−1^ (νas(COO-) in carboxylate), 1250 cm^−1^ (ν(C-O) in carboxylate), and 1642 cm^−1^ (ν(C=C) in backbone). These characteristic peaks are commonly observed in the spectra of CLs and CLCs, showing that the intact moieties of CA were well preserved in the hybrids. Furthermore, the spectrum of CAD revealed a peak at 1681 cm^−1^, which was derived from the ν(C=O) of the aldehyde functional group (Figure 4e) and was observed in the CLCs, suggesting the co-existence of both CA and CAD in the CLC hybrids. It was noteworthy that the ionic properties of CA in both the CL and CLC hybrids were fairly high to elucidate its strong electrostatic attraction toward the positive LDH layer. According to Chen et al.’s report [17], the percentage of ionic bond (PIB) could be calculated by [ν(COOH) − ν(COO-)]/[ν(COOH) − ν(COONa)]. The calculated PIB values of the cinnamic acid moiety in each hybrid were 1.07 (CL1), 1.09 (CL2), 1.02 (CL3), 1.06 (CLC1), 0.92 (CLC2), and 1.03 (CLC3), suggesting that the carboxylate moieties were highly ionic regardless of the Zn/Al ratio or the intercalated chemical species. It was therefore concluded that the organic moieties were well preserved in the interlayer space and were strongly bound to the LDH layer through electrostatic interaction.

### 2.3. Differential Scanning Calorimetry

In order to verify intermolecular interaction, possibly through π-π interaction in CLs, we carried out differential scanning calorimetry (DSC) under a N_2_ atmosphere (Figure 5). The DSC diagram for CA only showed a sharp endothermic peak at 133 °C, which corresponded to the melting point reported in the literature [18]. The calculated endothermic energy was ~2.9 kcal/mol, which was similar to the previously calculated π-π stabilization energy in the benzene moiety (~2.6 kcal/mol). The DSC diagrams of CLs showed three major endothermic peaks (below 100 °C, between 100 and 150 °C, and over 150 °C, respectively). The first peak was attributed to the evaporation of water, which was loosely bound at the surface of LDH. The second peak, which appeared at around 130 °C, was attributed to the π-π interaction among CA moieties. As the CA molecules are located in a random manner rather than being packed in an ordered array in the LDH gallery space as in CA crystal, the endothermic peak for π-π interaction would be broadened compared with CA alone. The last peak, above 150 °C, was due to the evaporation of water molecules in the interlayer space of LDH. It seems that CL1 had the largest amount of water, as the peak area above 150 °C was the biggest for CL1. It is not clearly addressed at this point why CL1 had the largest amount of interlayer water; however, we could suggest a possible explanation. The cinnamic acid moiety, which was hydrophilic due to the carboxylate moiety, could be easily hydrated. As CL1 would have the largest amount of cinnamic acid in the interlayer space among the three hybrids, the degree of hydration would also be the highest in CL1. We would like to emphasize, however, that the degree of hydration did not influence the stabilization of the CA moiety or the molecular arrangement in the CL hybrids.

### 2.4. Diffuse Reflectance Solid-State UV-Vis Spectroscopy

The intermolecular interactions in CLs and CLCs were also investigated with UV-vis diffuse reflectance spectra (Figure 6). The spectrum of CA showed two major bands at around 240 nm and 330 nm. The former band was attributed to a mixture of n→π* and π→π* transitions consisting of charge transfer from the aliphatic chain to the benzene ring. The latter band was due to the π→π* transition, which was related to the withdrawal of electronic charge density from the benzene ring to the aliphatic chain [15]. We could observe both peaks in the CL hybrids and could expect that the molecular arrangement of CA in the LDH interlayer space was fairly similar to that in CA powder, revealing the π→π interaction in CL. Nevertheless, the two peaks were slightly shifted; the peak at 240 nm red-shifted, and the peak at 330 nm blue-shifted. Both the lowering in charge transfer energy (the peak at around 240 nm) and the enhancement of charge density withdrawal (the peak at around 330 nm) could be attributed to the electron-rich character of the carboxylate moiety in the interlayer space of LDH. On the other hand, the spectra of all the CLC hybrids showed a tailing effect toward a higher wavelength in the charge density withdrawal band. Molecular packing among benzene moieties contributed to the delocalization of the π electron, resulting in a decrease in excitation energy, i.e., a red shift in the absorption band. It was previously reported that the π-π interaction of benzene-containing molecules increases the domain of conjugation, giving rise to the red shift in the absorption band [19]. The molecular packing in the interlayer space of LDH became dense upon CAD incorporation, as suggested in Figure 6. The delocalization of the π electron was more facilitated upon incorporation of CAD, giving rise to the stabilization of the CA-CAD composite in LDH. By this means, both CA and CAD could be well preserved in the gallery space of LDH even though the d-spacing expanded beyond the two-fold dimension of the CA molecule.

### 2.5. Particle Morphology of Hybrids

The overall morphologies of the CL and CLC hybrids were examined with scanning electron microscopy (SEM). As shown in Figure 7, all the CL and CLC hybrids exhibited an agglomeration of wavy sheets. As the host LDH is a stack of two-dimensional nanosheets, LDH particles tend to grow in a plate-like shape with a high diameter-to-thickness ratio. The agglomeration of LDH is mediated in an edge-to-edge, face-to-edge, or face-to-face manner through the unsaturated coordination site and the LDH layer [20], and the degree of agglomeration becomes more significant when organic molecules are intercalated. However, the agglomeration is a reversible phenomenon; the particles could be separated upon appropriate solvent treatment. The LDH particles containing small and tight inorganic acids such as carbonate and nitrate tend to have rigid and plate-like morphology [21]; on the other hand, the LDHs with relatively flexible organic molecules in their interlayer often exhibit bending of particles, giving rise to wavy morphology [22], as shown in CL and CLC. Although it is not easy to exactly define the size of primary particles from the agglomerated image, we could observe several particles with distinguishable particle dimensions, as denoted by the ~100 nm marker in the white circles in Figure 7. It is worth noting that the particle sizes of all the CL and CLC hybrids were comparable. First of all, the SEM results suggested that the coprecipitation of CL with different Zn/Al ratios did not alter the overall particle size. In other words, the crystal growth of ZnAl-LDH was not significantly affected by the Zn/Al ratio from 2/1 (CL1) to 7/1 (CL3). Second, the comparable size between CL and CLC represented that the CAD incorporation reaction occurred topotactically; there was net lattice expansion along the crystallographic c-axis without alteration in the ab-plane structure, as reported before [23]. From the molecular dimensions and Monte Carlo simulations, one CA or CAD molecule requires a cross-sectional area with a 0.7 nm diameter (Appendix A). Considering the average diameter of LDH (~100 nm) and the molecular dimensions, it can be suggested that there exist approximately 20,000 units of CA or CAD molecules in one interlayer space, giving rise to sufficiently strong intermolecular interactions through π-π interaction.

### 2.6. Time-Dependent CA Release in Aqueous Media

Intermolecular interaction, CA-CA or CA-CAD-CA, in addition to the electrostatic interaction between CA and the LDH layer, was expected to influence the release property of incorporated CA. As shown in Figure 8, we carried out CA release from either CL or CLC in deionized water.

The first step in the release analyses is to compare the time-dependent CA release profile among CL hybrids with different Zn/Al ratios. CA release was the most suppressed in CL1, which has the lowest Zn/Al ratio, or, in other words, the highest layer charge density. The CA release from the CL hybrid is mediated by the competition between electrostatic attraction and the hydration of CA by water. As the hydration enthalpy is always constant, the release rate would be governed by the electrostatic attraction. Although there exists only one (+)-(-) interaction between LDH and CA, the high positive charge density of CL1 could prevent a CA molecule from easily escaping from the gallery space of LDH. In this regard, CL2 and CL3, which had fairly low charge densities in the LDH layer, would show more CA release.

The second point to consider is the different release profiles between CL and CLC depending on the Zn/Al ratio. CLC2 and CLC3 showed suppressed CA release compared with CL. Although the electrostatic interaction between CA and LDH is not so strong in CLC2 and CLC3, the additional stabilization obtained by the CAD moiety through π-π interaction hindered CA from escaping from the gallery space of LDH. Moreover, CAD has hydrophobic properties, and therefore, water molecule access around it was hindered, reducing the chances of hydration for the CA moiety attracted by CAD. The results strongly suggested that the additional incorporation of CAD could stabilize CA molecules in the gallery space and that the π-π mediator could influence the release property of intercalated molecules. As it was previously reported that the intact layer structure of LDH is well preserved after payload release [24], we could expect that the existence of CAD would not alter the structure of LDH after release but modify the release kinetics of the cinnamic acid moiety.

The CA release in CLC1 was more accelerated compared with CL1, which was the reverse phenomenon in CLC2 and CLC3. Although the phenomenon is opposite to what we expected from the strong π-π interaction, we could suggest a plausible explanation for this finding based on the access of water molecules to the expanded interlayer space. The amount of CAD is the lowest in CLC1 due to the high amount of CA. At the same time, the interlayer distance of CLC1 expanded, like in CLC1 or CLC2. The incorporation of CAD has two controversial effects on CA release: (i) slowing down the release by providing π-π interaction and (ii) accelerating the release by expanding interlayer space to accommodate more solvent molecules. The accelerating effect of CAD in CLC1 would be higher than the slowing effect, despite the small amount, compared with CLC2 and CLC3, due to the same lattice expansion. In this manner, the CA molecules in the CLC1 hybrid would have more chances to be hydrated than in CL1, resulting in a fast release.

Based on the release results in neutral pH conditions and a previous report [24], we could expect that the release of CA in an acidic pH would be more than 40% within 24 h. As the cinnamic acid-stabilized LDH can be utilized in cosmetics for preserving the CA moiety during product processing, we could assume that the existence of a π-π mediator like CAD prevents CA from being released unexpectedly during manufacturing.

## 3. Materials and Methods

### 3.1. Materials

Zinc nitrate hexahydrate (Zn(NO_3_)_2_·6H_2_O; 98%), aluminum nitrate nonahydrate (Al(NO_3_)_3_·9H_2_O; 98%), trans-cinnamic acid (CA, C_9_H_8_O_2_; 97%), trans-cinnamaldehyde (CAD, C_9_H_8_O; 99%), and potassium bromide (KBr; 99%) were purchased from Sigma-Aldrich, St. Louis, MO, USA. Sodium hydroxide (NaOH; 97%), ethyl alcohol (C_2_H_5_OH; 99%), and hydrochloric acid (HCl; 36%) were obtained from Daejung Chemicals & Metals, Seoul, Republic of Korea. Acetic acid (CH_3_COOH; 99%) and acetonitrile were purchased from Junsei, Tokyo, Japan, and Honeywell Burdick & Jackson, Muskegon, MI, USA, respectively. The deionized water was obtained from a water purification system (Human Power II+ Water Purification System, Human Corporation, Seoul, Republic of Korea) with a resistivity of 18.3 MΩ/cm.

### 3.2. Incorporation of Cinnamic Acid and Cinnamaldehyde

The CA molecules were incorporated into LDH by coprecipitation method. Powdered CA (0.006 mol) was dispersed in decarbonated water (100 mL), and an equivalent amount of sodium hydroxide (0.006 mol) was added with vigorous stirring under N_2_ atmosphere to obtain carboxylate. Then, the mixed metal solution (Zn^2+^/Al^3+^) and alkaline solution (0.5 M of NaOH) were sequentially added to the CA solution until the pH reached ~8.0. The white suspension was aged for 24 h with vigorous stirring. The product was collected by centrifugation (10,000 rpm, 3 min) and thoroughly washed with decarbonated water. CA-incorporated LDH is named CL. Different Zn^2+^/Al^3+^ ratios were tried to control the charge density of LDH. Numbers 1, 2, and 3 were added after “CL”, indicating a Zn^2+^/Al^3+^ ratio of 2/1, 4/1, and 7/1, respectively. The CAD moiety was additionally incorporated into the interlayer space of CLs (CLCs). Typically, CLs were dispersed in EtOH, and the suspensions were stirred vigorously at 65 °C under N_2_ atmosphere. Then, liquid CAD was added to the suspension. After 24 h of reaction, the products were collected by centrifugation at 10,000 rpm for 3 min and washed with fresh EtOH. For comparison, ZnAl-LDH with nitrate anions was synthesized as follows: The mixed metal solution (Zn^2+^/Al^3+^ = 4/1) and alkaline solution (0.5 M of NaOH/0.15 M of NaNO_3_) were prepared. The metal solution was titrated with an alkaline solution until the pH reached ~8.0 under vigorous stirring. The white suspension was aged for 24 h at 65 °C under N_2_ atmosphere. The obtained precipitate was isolated by centrifugation (10,000 rpm, 3 min) and washed with decarbonated water several times. All the LDH samples were lyophilized after washing for further characterization.

### 3.3. Characterization of CLs and CLCs

The crystal structures of the prepared samples were identified using an X-ray diffractometer (XRD) with a Bruker AXS D2 phaser (LYNXEYE detector) using Ni-filtered Cu-Kα radiation (λ = 1.5406 Å) with 1 mm of air-scattering slit and 0.1 mm of equatorial slit. The diffractograms were obtained with time step increments of 0.02° at a scanning rate of 0.5 s/step. Fourier transformed infrared (FT-IR; Spectrum one B.v5.0, Perkin Elmer, Waltham, MA, USA) spectra were obtained with the conventional KBr pellet method. FT-IR spectra were recorded from 450 to 4000 cm^−1^.

Intermolecular interactions were investigated by utilizing a differential scanning calorimeter (DSC; Q200, TA Instrument, New Castle, DE, USA) and UV-vis spectrometer (EVOLUTION 220, Thermo Scientific, Waltham, MA, USA). DSC measurements were carried out over a temperature range from 25 °C to 250 °C with a heating rate of 2 °C/min under flowing N_2_ gas. Solid-state UV-vis spectra of powdery CLs, CLCs, and CA were only obtained in a range from 190 nm to 500 nm with BaSO_4_ background.

Chemical formulae were determined by using an inductively coupled plasma-optical emission spectrometer (ICP-OES; OPTIMA 7300 DV, Perkin Elmer, Waltham, MA, USA) and high-performance liquid chromatography (HPLC; YL9100, Younglin, Anyang-si, Korea). For ICP-OES pre-treatment, powdered samples were thoroughly digested in hydrochloric acid and evaporated after digestion. The remnant was diluted with deionized water to reach an adequate metal concentration (10–100 ppm). The content of CA and CAD was quantified by HPLC equipped with a C18 column (ZOBAX Eclipse, 4.6 × 150 mm, Agilent, Santa Clara, CA, USA). The mobile phase was 0.5% aqueous glacial acetic acid/water/acetonitrile (25:30:45) at a flow rate of 1.0 mL/min. The UV absorption detector and column temperature were set at 276 nm and 25 °C, respectively. Before the evaluation, the powdery samples were totally dissolved in phosphate buffer solution (pH ~2) by stirring for 10 min and sonicating for 10 min to digest the LDH lattices.

Size and morphology were visualized by scanning electron microscopy (SEM; FEI, QUANTA 250 FEG, Philips, Amsterdam, The Netherlands). The SEM specimens were prepared as follows: powdered samples were spread on carbon tape, and their surfaces were coated with Pt/Pd plasma sputtering for 50 s. The SEM images were collected at 30 kV of accelerated electron beam.

### 3.4. Time-Dependent CA Release Profiles

The time-dependent release profiles of CA from CLs and CLCs were evaluated at 25 °C in deionized water. Typically, 50 mg of each powder sample was located in 100 mL of deionized water under moderate stirring. Aliquots (1 mL) were collected at designated time points, and the cumulative CA release was determined through high-performance liquid chromatography (HPLC; YL9100, Younglin) equipped with a C18 column (ZOBAX Eclipse, 4.6 × 150 mm, Agilent). The mobile phase was 0.5% aqueous glacial acetic acid/water/acetonitrile (25:30:45) at a flow rate of 1.0 mL/min. The UV absorption detector and column temperature were set at 276 nm and 25 °C, respectively.

### 3.5. Computational Simulations

In order to probe the potential adsorption sites and the plausible configuration of the guest cinnamic acid ions and cinnamaldehyde molecules, Monte Carlo calculations were performed in three different ZnAl structures. Using experimental data, three different ZnAl structures were built to reproduce the Zn/Al ratio in order to obtain Zn_2_Al, Zn_4_Al, and Zn_7_Al with the corresponding experimental interlayer space openings. In order to perform Monte Carlo calculations, a classical force field containing partial charges for the solids and for the guest molecules/ions is required. Thus, from these structures, the partial charges were calculated using the electronegativity equalization formalism available in Materials Studio (qEq charges), and the results are reported in Table 1.

Regarding cinnamic acid ions and cinnamaldehyde, the partial charges were calculated using DFT calculations with DMol3 from the Materials Studio package, using GGA/PW91 as a functional and DNP basis set after a geometry optimization step. From the DFT calculations, the Electrostatic potential model (ESP) was used to extract partial charges, and the results are reported in Figure 9.

Each atom of the guest and host was then considered to calculate the adsorbate–adsorbent interactions, corresponding to the sum of repulsion–dispersion 12-6 Lennard-Jones (LJ) potential combined with a coulombic potential. The LJ parameters for each atom were taken from the Universal Force Field (UFF), and they were combined following the Lorentz–Berthelot rules, while the electrostatic part was calculated using the Ewald summation by considering mulicells (formed by 9 × 3 × 1 unit cells). The Ewald summation was used for simulating the electrostatic interactions with an accuracy fixed at 0.001 kcal mol^−1^ and a buffer width of 0.05 nm.

Using this force field, Monte Carlo simulations at ambient temperature were performed using the SORPTION module available in Materials Studio. Calculations were performed by saturating the layer charge in the presence of cinnamic acid ions, which were intercalated in the interlayer spaces. It should be mentioned that the interlayer space opening obtained experimentally was large enough to allow the saturation of the layer charge by using only the cinnamic acid ions.

In a typical run of computations, the simulation box corresponded to 27 unit cells in order to use an LJ cut-off equal to 1.2 nm. The simulations were performed using 2 × 10^6^ Monte Carlo steps for both equilibration and production steps after a loading procedure fixed at 100 × 10^3^ steps to insert the cinnamic acid ions. The framework and the guest ions and molecules were kept rigid during the whole adsorption process.

Monte Carlo calculations were performed with different loadings: for Zn_2_Al, 54 cinnamic acid ions and 20 cinnamaldehyde molecules were adsorbed; for Zn_4_Al, 32 cinnamic acid ions and 20 cinnamaldehyde molecules; and for Zn_7_Al, 20 cinnamic acid ions and 20 cinnamaldehyde molecules. The number of cinnamaldehyde molecules is chosen to observe interactions with the LDH framework, cinnamic acid ions, and other cinnamaldehyde molecules.

From these calculations, it was possible to extract the adsorption enthalpy for CAD molecules and CA ions in the different LDHs as a function of the chemical composition, as well as the main plausible interactions elucidated from a microscopic point of view. The plausible configurations correspond to the lowest-energy configurations obtained after Monte Carlo calculations.

## 4. Conclusions

The molecular arrangement and stabilization of CA were achieved by utilizing a two-dimensional substrate, LDH, and an additive, CAD, to enhance intermolecular interactions. The anionic CA molecules were successfully stabilized in the gallery space of LDH through electrostatic interaction, and the molecules were arranged in an inter-digitated manner to maximize intermolecular interaction. Different from other organic–LDH materials, CL seemed to utilize both electrostatic interaction and intermolecular π-π interactions for the molecular arrangement of cinnamic acid. In order to introduce additional stabilization to the CA arrangement, a neutral derivative of CA, cinnamaldehyde, was incorporated into the gallery space of CL. CAD introduction expanded the lattice along the c-axis through the additional intermolecular interactions between CA and CAD, which was cross-confirmed by the Monte Carlo simulation. The strong intermolecular interaction through π-π stacking as well as the electrostatic stabilization were examined by FT-IR, UV-vis spectroscopy, and DSC. Finally, the time-dependent CA release from either the CL or CLC hybrids was evaluated. Generally, the release was strongly governed by the electrostatic interaction between LDH and CA; a higher positive charge density of LDH slowed down CA release. However, the addition of CAD modified the release kinetics; the more CAD existed in the CLC hybrids, the more slowly the CA molecules were released. However, the small amount of CAD incorporation accelerated CA release due to lattice pre-expansion. We could therefore conclude that the incorporation of a π-π mediator in phenolic acid-intercalated LDH could slow down or accelerate the release of payloads, depending on the amount of mediator. Taking into account the potential application of phenolic acids in cosmetics or food ingredients, the utilization of π-π interactions with neutral derivatives could enhance the stability of phenolic acid and provide ease of handling for the production process.

## Figures and Tables

**Figure 1 ijms-25-04506-f001:**
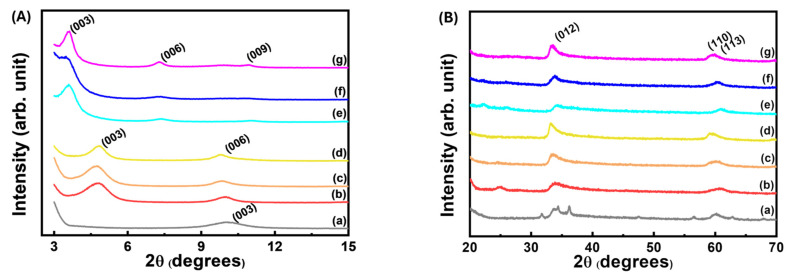
X-ray diffraction patterns in the high-angle region for (a) ZnAl-${NO}_{3}^{-}$-LDH, (b) CL1, (c) CL2, (d) CL3, (e) CLC1, (f) CLC2, and (g) CLC3. (**A**) low 2θ region and (**B**) high 2θ region.

**Figure 2 ijms-25-04506-f002:**
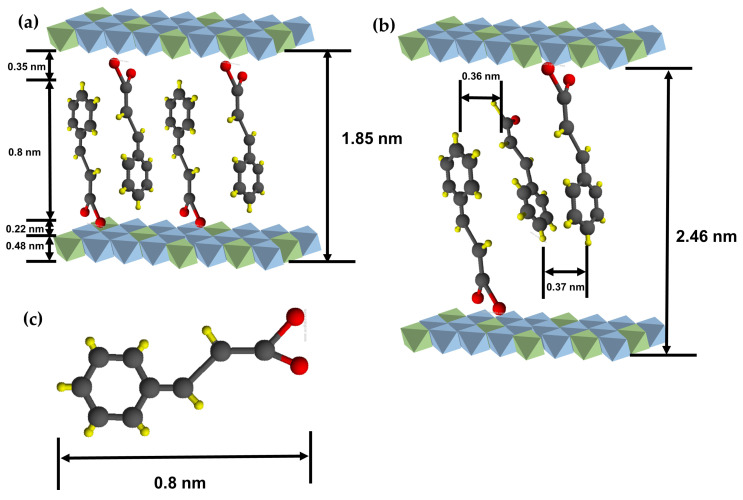
Schematic diagrams of interlayer molecular arrangements: (**a**) CL and (**b**) CLC. (**c**) Dimension of a cinnamic acid molecule. The color of atom representation is as follows: H (yellow), C (black), O (red), Al (green), Zn (blue).

**Figure 3 ijms-25-04506-f003:**
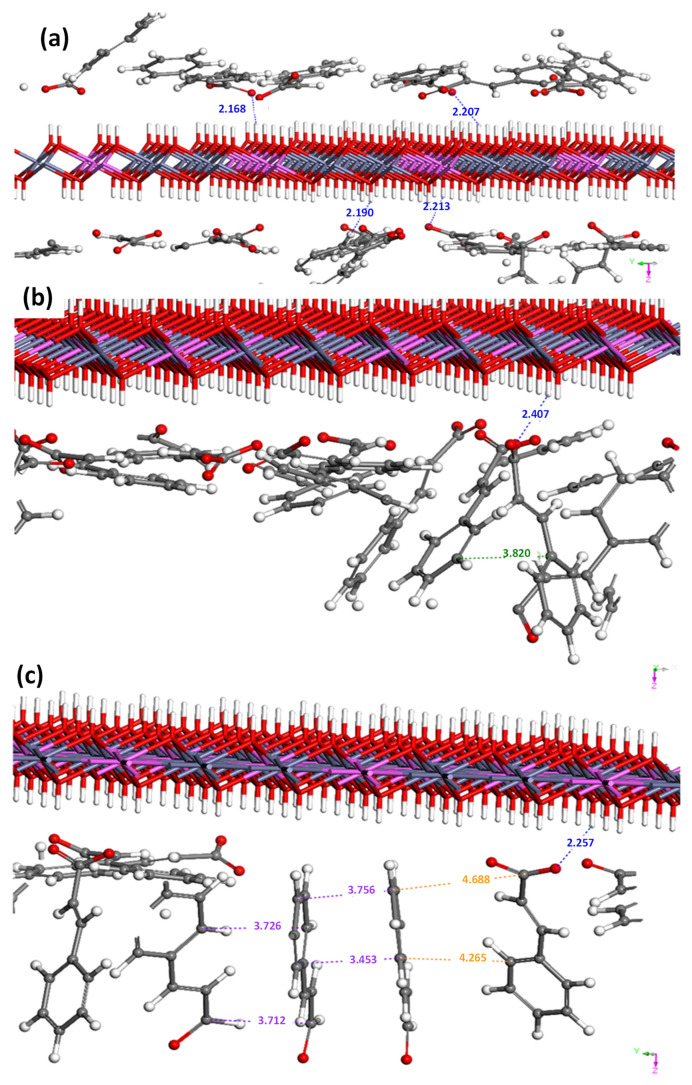
Plausible configurations for CA and CAD on the LDH surface extracted from Monte Carlo simulations. The main interactions between CA and the LDH framework, CA-CA, CA-CAD, and CAD-CAD are in blue, green, orange, and purple, respectively. The color of atom representation is classical: H (white), C (red), Zn (dark gray), and Al (pink). (**a**) Interactions between CA and the LDH framework, (**b**) Interactions between CA-CA and CA-LDH (**c**) Interactions between CA-CA, CA-CAD, CAD-CAD and CA-LDH.

**Figure 4 ijms-25-04506-f004:**
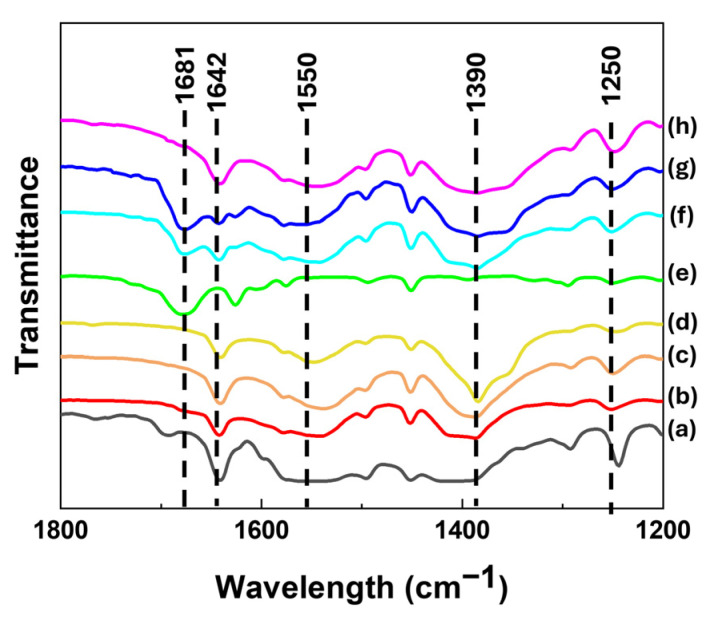
Fourier Transform infrared spectra of (a) anionic form of CA, (b) CL1, (c) CL2, (d) CL3, (e) CAD, (f) CLC1, (g) CLC2, and (h) CLC3.

**Figure 5 ijms-25-04506-f005:**
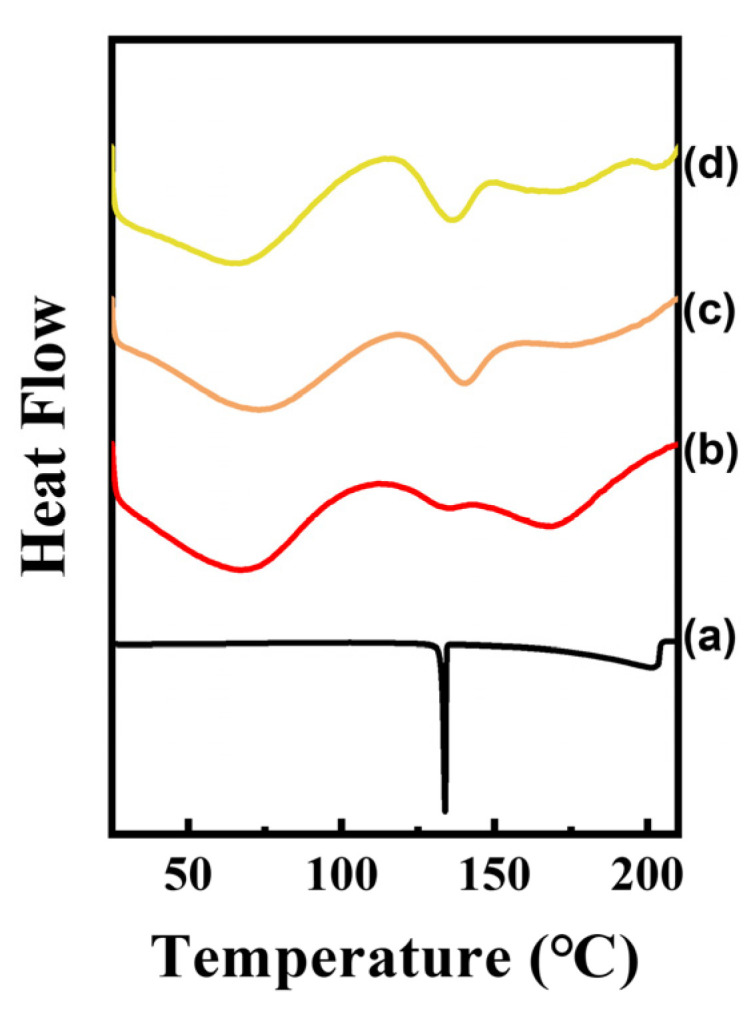
Differential scanning calorimetry (DSC) results for (a) CA, (b) CL1, (c) CL2, and (d) CL3. The DSC diagram was drawn with offsets to avoid overlapping curves.

**Figure 6 ijms-25-04506-f006:**
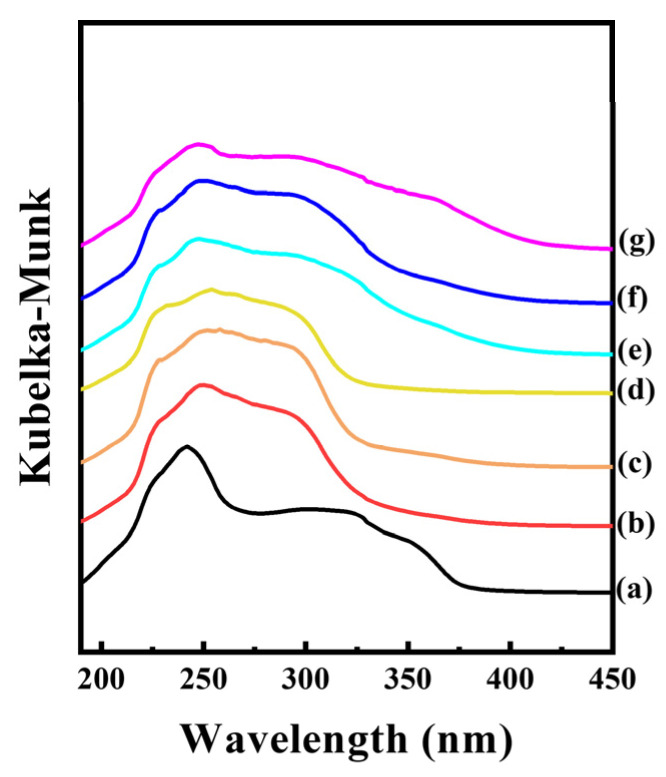
Solid-state diffuse reflectance spectra of (a) CA, (b) CL1, (c) CL2, (d) CL3, (e) CLC1, (f) CLC2, and (g) CLC3. For effective comparison, the peak maximums were normalized to have a similar value.

**Figure 7 ijms-25-04506-f007:**
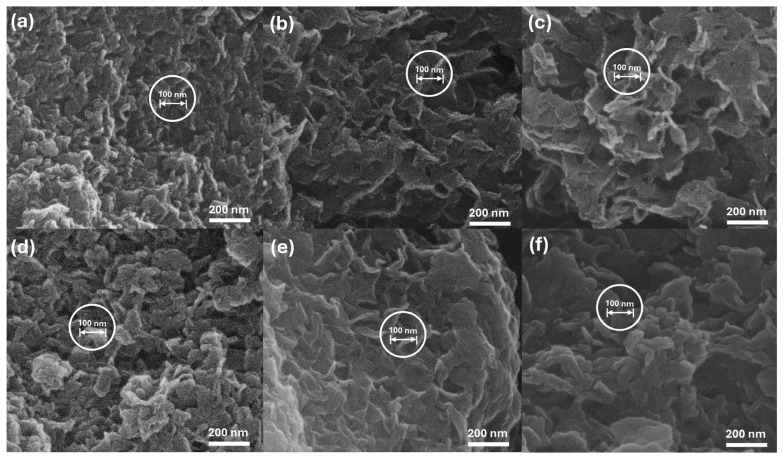
Scanning electron microscopic images of (**a**) CL1, (**b**) CL2, (**c**) CL3, (**d**) CLC1, (**e**) CLC2, and (**f**) CLC3.

**Figure 8 ijms-25-04506-f008:**
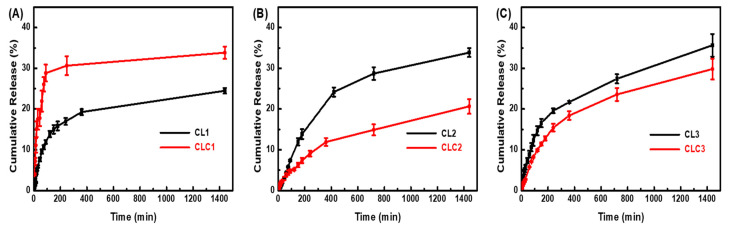
Time-dependent release profile of CA in deionized water for (**A**) CL1, CLC1; (**B**) CL2, CLC2; and (**C**) CL3, CLC3.

**Figure 9 ijms-25-04506-f009:**
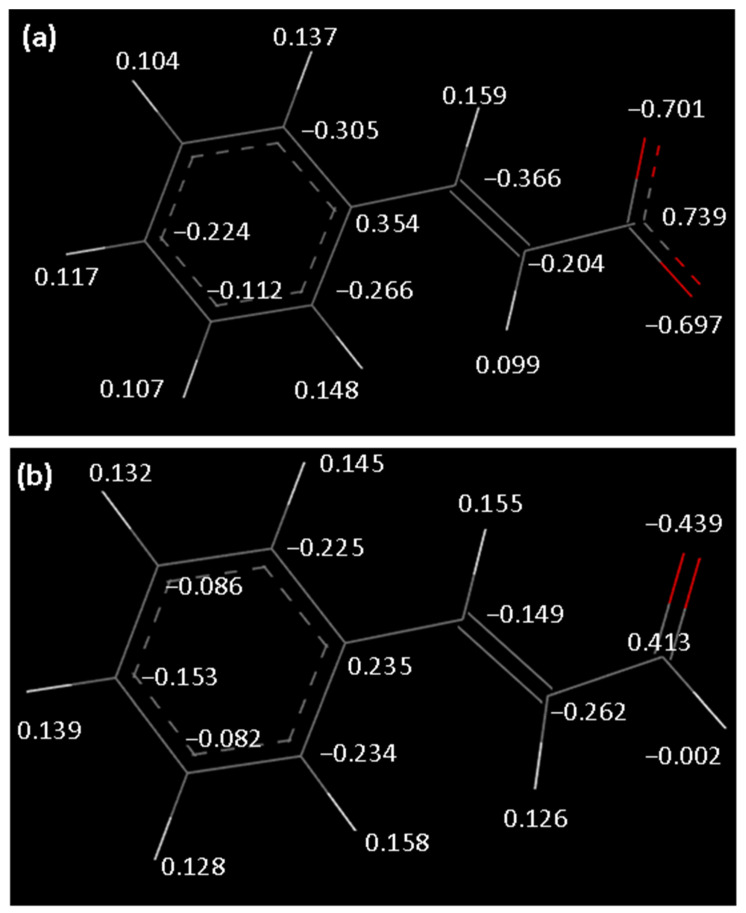
Partial charges extracted for DFT calculations for (**a**) CA ion and (**b**) CAD molecule.

**Table 1 ijms-25-04506-t001:** Partial charges for the atoms of the LDH framework obtained from the qEq method available in Materials Studio.

	H	O	Zn	Al
Zn_7_Al	0.24546	−0.975205	1.4148	1.79579
Zn_4_Al	0.255336	−0.965329	1.42467	1.800567
Zn_2_Al	0.273443	−0.947222	1.44278	1.82378

## Data Availability

All data generated or analyzed during this study are included in this article.

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
