# Peer review of "Controlled Molecular Arrangement of Cinnamic Acid in Layered Double Hydroxide through pi-pi Interaction for Controlled Release"

_ijms, 2024, doi:10.3390/ijms25084506_

Round 1

Reviewer 1 Report

Comments and Suggestions for Authors

Utilizing experiments and computational simulations, cinnamic acid and its derivatives enhance the interlayer stability of LDH through pi-pi interaction to delay the release kinetics. The paper contains some interesting results, but the characterization is not enough to support its conclusions. Therefore, I suggest this manuscript should be put forward for major revision.

Here are some specific comments/suggestions:

1.      (Page 3 line 104) “Comparing with the (003) peak of pristine ZnAl-NO3-LDH corresponding to d-spacing of 0.08 nm, the (003) peaks of all the CLs shifted to lower 2θ region.”Explain how the crystal plane spacing is measured.

2.      (Page 4 line 128)“The theoretical demanded area of positive charge in LDH is 0.25, 0.42 and 0.67 nm2 for 2/1, 4/1 and 7/1 ratio (CL1, CL2, and CL3, respectively).”What is the theoretical demand area of positive charge and how to calculate it?

3.      (Page 4 line 138)“Upon additional incorporation of CAD, the lattice of CL again expanded along crystallographic c-axis showing final d-spacing of 24.6 (Fig. 1).” I propose to unify the distance units in the whole paper and explain which peak position in Figure 1 shows that the crystal face spacing d is 2.46nm.

4.      (Page 6 line 152)How to select the three states in Figure 3 (A), (B), and (c).

5.      (Page 7 line 156)“The spectrum of anionic CA (Fig. 4(a)), which was obtained by deprotonation of CA, showed characteristic peaks at ~1390, 1550, 1250, and 1642 cm-1 attributed to νs(COO-), νas(COO-), ν(C-O) and backbone ν(C=C), respectively.”What characteristic peak does 1250 refer to?

6.      (Page 7 line 165)“The calculated PIB values of CA were almost 1 for all the hybrids, suggesting that the carboxylate moieties were highly ionic regardless of Zn/Al ratio or intercalated chemical species.”How is the percentage of CA ionic bond 1 calculated?

7.      (Page 8 line 186)“Differential scanning calorimetry (DSC) result of (a) CA, (b) CL1, (c) CL2, (d) CL3. Both ends of the curves were fitted straight with Origin®2018. For efficient comparison, the peak minimums were normalized to have similar values.”It is recommended to put the whole word value together.

8.      (Page 9 line 201)“Molecular packing among benzene moieties contributed to the delocalization of the electron, resulting in the decrease of excitation energy.”Should the statement of Π electron delocalization be illustrated by the DFT calculation? How does ultraviolet spectroscopy illustrate electron delocalization?

9.      (Page 10 line 213)“As the host LDH is a stacking of 2-dimensional nanosheets, the LDH particles tend to grow in plate-like shape with high diameter-to-thickness ratio.”What's the difference between rigid plate-like and plate-like?

10.   (Page 10 line 227)“Considering the molecular dimension of CA or CAD molecules, the average diameter of ~ 100 nm suggested that there existed approximately 15,000 units of CA or CAD molecules in one interlayer space, giving rise to sufficiently strong intermolecular interaction through π-π interaction.”Please explain why there are 15,000 cases for 100nm CL. How is it calculated? 

11.   (Page 11 line 258)“This would be attributed to the lower amount of CAD in CLC3. Due to the high molecular packing in CL3, there would be less CAD incorporation in the CLC3 hybrid. Nevertheless, the interlayer space was expanded by the CAD incorporation.”I think there is a problem with the logic of this statement and there is no good explanation as to why the first figure in Figure 8 shows the opposite phenomenon.

12.   (Page 16 line 371)“In a typical run of computations, the simulation box corresponded to 27 unit cells to use an LJ cut-off equal to 1.2 nm.”Why use an LJ cut-off equal to 1.2nm?

13.   Some of the images in this article suggest colorization, the colors are too simple, and some images are not clear enough, Some of the formatting of the article is wrong, as follows:

1)       The curves in Figure 1Figure4Figure5Figure6need to be colored.

2)       The clarity of Figure 7 is too low.

3)       The subheading in the Materials and Methods section is not numbered correctly

Author Response

Response to reviewer 1

Utilizing experiments and computational simulations, cinnamic acid and its derivatives enhance the interlayer stability of LDH through pi-pi interaction to delay the release kinetics. The paper contains some interesting results, but the characterization is not enough to support its conclusions. Therefore, I suggest this manuscript should be put forward for major revision.

Here are some specific comments/suggestions:

  1. (Page 3 line 104) “Comparing with the (003) peak of pristine ZnAl-NO3-LDH corresponding to d-spacing of 0.08 nm, the (003) peaks of all the CLs shifted to lower 2θ region.” Explain how the crystal plane spacing is measured.

=> We appreciate the reviewer’s specific comment. We should have included detailed method for the crystal plane calculation. In this study, Bragg’s equation which is defined as nλ=2dsinθ (d: d-spacing corresponding to a specific lattice plane, n: integer, λ: wavelength of X-ray, 0.154 nm for Cu Kα in this study, θ: Bragg angle)  was utilized applied for (003) peak. As the layer stacking of LDH is in rhombohedral manner, three layers make one unit cell in the crystallographic c-axis. Thus, the d-value calculated from (003) peak could be directly utilized as the d-spacing of layer stacking. We revised the manuscript accordingly.

Added : Page 3 line 117) In order to calculate d-spacing of along crystallographic c-axis, Bragg’s equation which is defined as nλ=2dsinθ (d: d-spacing corresponding to a specific lattice plane; n: integer, 1 was applied in this study; lambda: wavelength of X-ray, 0.154 nm for Cu Kα in this study; θ: Bragg angle) was applied for (003) peak, as the layer stacking of LDH is in rhombohedral manner – three layers make one unit cell. 

  1. (Page 4 line 128)“The theoretical demanded area of positive charge in LDH is 0.25, 0.42 and 0.67 nm2 for 2/1, 4/1 and 7/1 ratio (CL1, CL2, and CL3, respectively).”What is the theoretical demand area of positive charge and how to calculate it?

=> We appreciate the reviewer’s advice to enhance comprehension of readers. As shown in the figure below, the red hexagon contains two Zn2+ ions and one Al3+ ion, representing a unit of Zn2Al(OH)6+ of an LDH layer (Zn/Al ratio of 2/1). Taking into account the cell parameter a=0.305 nm, the area of the hexagon, which contains one positive charge, is approximately 0.25 nm2. The demanded area for ZnAl-LDH with another Zn/Al ratio could also be calculated based on this schematic diagram. The demanded areas of Zn/Al=4/1 and 7/1 are 5/3 and 8/3 times larger than that of Zn/Al=2/1, giving rise to the area value of 0.42 nm2 and 0.67 nm2. In order to help the readers’ comprehension we added Fig. S1 and denoted the part in the manuscript as follows.

Original Page 4 line 128)  “The theoretical demanded area of positive charge in LDH is 0.25, 0.42 and 0.67 nm2 for 2/1, 4/1 and 7/1 ratio (CL1, CL2, and CL3, respectively).”

Revised Page 4 line 147) The theoretical demanded area of positive charge in LDH can be calculated based on the unit cell diagram along ab-plane as shown in Fig. S1. The LDH with Zn/Al ratio 2/1 produces one positive charge in a 0.25 nm2 hexagon (CL1). The demanded area increases 5/3 and 8/3, respectively for Zn/Al ratio 4/1 and 7/1, resulting in the demanded area value of 0.42 (CL2) and 0.67 nm2 (CL3)

  1. (Page 4 line 138)“Upon additional incorporation of CAD, the lattice of CL again expanded along crystallographic c-axis showing final d-spacing of 24.6 Å (Fig. 1).” I propose to unify the distance units in the whole paper and explain which peak position in Figure 1 shows that the crystal face spacing d is 2.46nm.

=> We appreciate the reviewer’s advice. We unified the distance unit into nm throughout the manuscript.

  1. (Page 6 line 152)How to select the three states in Figure 3 (A), (B), and (c).

=> The configurations selected for Figure 3 correspond to the most plausible configurations extracted from Monte Carlo simulations. They are determined by the lowest energy and therefore the most probable and stable configurations. In the different figures, we focus on the main interactions that we wanted to make as evidence.

We have added a sentence in the Materials and Methods part:

Added: Page 19 line 443) The plausible configurations correspond to the lowest energy configurations obtained after Monte Carlo calculations.

  1. (Page 7 line 156)“The spectrum of anionic CA (Fig. 4(a)), which was obtained by deprotonation of CA, showed characteristic peaks at ~1390, 1550, 1250, and 1642 cm-1 attributed to νs(COO-), νas(COO-), ν(C-O) and backbone ν(C=C), respectively.” What characteristic peak does 1250 refer to?

=> As commented by the reviewer, the sentence in the original version is not clear. In order to enhance the legibility, we revised the sentence as follows.

Original version: Page 7 line 156) The spectrum of anionic CA (Fig. 4(a)), which was obtained by deprotonation of CA, showed characteristic peaks at ~1390, 1550, 1250, and 1642 cm-1 attributed to νs(COO-), νas(COO-), ν(C-O) and backbone ν(C=C), respectively.

Revised version: Page 8 line 176) The spectrum of anionic CA (Fig. 4(a)), which was obtained by deprotonation of CA, showed characteristic peaks at ~1390 cm-1 (νs(COO-) in carboxylate), 1550 cm-1 (νas(COO-) in carboxylate), 1250 cm-1 (ν(C-O) in carboxylate), and 1642 cm-1 (ν(C=C) in backbone).

  1. (Page 7 line 165)“The calculated PIB values of CA were almost 1 for all the hybrids, suggesting that the carboxylate moieties were highly ionic regardless of Zn/Al ratio or intercalated chemical species.” How is the percentage of CA ionic bond 1 calculated?

=> We appreciate the reviewer’s valuable comments. The percentage of ionic bond (PIB) was calculated for the cinnamic acid moiety in each hybrid. The terminology “CA” was not clear so that the readers may misunderstand.  We utilized “CA moiety in the hybrid” instead of “CA”. Furthermore, we represented the calculated PIB for each sample to clearly show that the value was almost 1 regardless of the sample. The manuscript was revised as follows.

Original: Page 7 line 165) The calculated PIB values of CA were almost 1 for all the hybrids, suggesting that the carboxylate moieties were highly ionic regardless of Zn/Al ratio or intercalated chemical species.

Revised: Page 8 line 187) The calculated PIB values of cinnamic acid moiety in each hybrid were 1.07 (CL1), 1.09 (CL2), 1.02 (CL3), 1.06 (CLC1), 0.92 (CLC2), and 1.03 (CLC3), suggesting that the carboxylate moieties were highly ionic regardless of Zn/Al ratio or intercalated chemical species.

  1. (Page 8 line 186)“Differential scanning calorimetry (DSC) result of (a) CA, (b) CL1, (c) CL2, (d) CL3. Both ends of the curves were fitted straight with Origin®2018. For efficient comparison, the peak minimums were normalized to have similar values.”It is recommended to put the whole word value together.

=> We deeply appreciate the reviewer’s comments. We are very sorry that the sentence “Both ends of the curves were fitted straight with Origin®2018. For efficient comparison, the peak minimums were normalized to have similar values” was totally our mistake. We have interpreted the DSC data many times and at some point, we tried to normalize the values of peak minimum; however, according to our raw data matching, the current DSC diagram was drawn without peak straightening nor normalization. We revised the figure caption as below.

Original version: Page 8 line 186) Differential scanning calorimetry (DSC) result of (a) CA, (b) CL1, (c) CL2, (d) CL3. Both ends of curves were fitted straight with Origin®2018. For efficient comparison, the peak minimums were normalized to have similar value.

Revised version: Page 10 line 227) Differential scanning calorimetry (DSC) result of (a) CA, (b) CL1, (c) CL2, (d) CL3. The DSC diagram was drawn with offsets to avoid overlapping of curves.//

  1. (Page 9 line 201)“Molecular packing among benzene moieties contributed to the delocalization of the electron, resulting in the decrease of excitation energy.”Should the statement of Π electron delocalization be illustrated by the DFT calculation? How does ultraviolet spectroscopy illustrate electron delocalization?

=> We appreciate the reviewer’s comment. The pi electron delocalization was interpreted from the peak shift in the solid state UV-vis spectrum. The absorption band of UV-vis spectra usually shifted toward higher wavelength when the molecules have additional conjugation effect. This phenomenon was previously reported in the covalent conjugation of folic acid with inorganic lattice [J.M. Oh, et al. Inorganic Drug-Delivery Nanovehicle Conjugated with Cancer-Cell-Specific Ligand, Advanced Functional Materials, 2009, 19, (10), 1617-1624] and the pi-pi stacking of naphthalimide dye [Cao. X, et al. Large red-shifted fluorescent emission via intermolecular π–π stacking in 4-ethynyl-1, 8-naphthalimide-based supramolecular assemblies, Langmuir, 2014, 30, (39), 11753-11760]. As we could observe slight shift of absorption band toward longer wavelength region, we could suggest the pi-pi interaction between cinnamic acid moieties inside the CL or CLC hybrids. To make this point clear, we revised the manuscript as follows.

Original Page 9 line 201)Molecular packing among benzene moieties contributed to the delocalization of the p electron, resulting in the decrease of excitation energy.

Revised Page 10 line 241) Molecular packing among benzene moieties contributed to the delocalization of the p electron, resulting in the decrease of excitation energy, i.e, red-shift in absorption band. It was previously reported that the p-p interaction of benzene-containing molecules increases the domain of conjugation, giving rise to the red-shift in the absorption band

  1. (Page 10 line 213)“As the host LDH is a stacking of 2-dimensional nanosheets, the LDH particles tend to grow in plate-like shape with high diameter-to-thickness ratio.”What's the difference between rigid plate-like and plate-like?

=> We apologize the review for our awkward demonstration. The expression “rigid plate-like” should be revised to “rigid and plate-like”. The LDH particles used to grow to have plate-like morphology. When a small anion is intercalated, the particle tends to have rigid shape, while the organic-intercalated LDH shows wavy shape. The expression “rigid and plate-like morphology” was used to show that the wavy and flexible nature of LDH plates with organic molecules. We revised the wording as follows.

Original Page 10 line 213) rigid plate-like morphology

Revised Page 12 line 262) rigid and plate-like morphology

  1. (Page 10 line 227)“Considering the molecular dimension of CA or CAD molecules, the average diameter of ~ 100 nm suggested that there existed approximately 15,000 units of CA or CAD molecules in one interlayer space, giving rise to sufficiently strong intermolecular interaction through π-π interaction.”Please explain why there are 15,000 cases for 100nm CL. How is it calculated? 

=> We agree with the reviewer that the detailed calculation should be addressed. Taking into account the molecular dimension and the intermolecular distance obtained from Monte Carlo simulation, we hypothesized that a cinnamic acid or cinnamaldehyde molecule requires cross-sectional area with diameter 0.7 nm. And the lateral dimension of one hybrid layer was approximated to 100 nm from the SEM measurement. Supposing that the CA and CAD molecules are closely packed along the LDH layers, maximum 20,000 molecules (calculation below) can be accommodated in one LDH layer.

Area of an LDH layer: 50 nm ´ 50 nm ´ pi ~ 7,850 nm2

Demanded area of a CA or CAD molecule: 0.35 nm ´ 0.35 nm ´ pi ~ 0.385 nm2

7,850/0.385 ~ 20,000 unit of CA(D) along a layer of LDH

In order to clarify this point, we added Fig. S2 in the supporting information and revised the manuscript accordingly.

Original Page 10 line 227) “Considering the molecular dimension of CA or CAD molecules, the average diameter of ~ 100 nm suggested that there existed approximately 15,000 units of CA or CAD molecules in one interlayer space, giving rise to sufficiently strong intermolecular interaction through π-π interaction.”

Revised Page 12 line 274) From the molecular dimension and Monte Carlo simulation, one CA or CAD molecule requires a cross-sectional area with 0.7 nm diameter (Fig. S2); considering the average diameter of LDH ~100 nm and the molecular dimension, it can be suggested that there existed approximately 20,000 units of CA or CAD molecules in one interlayer space, giving rise to sufficiently strong intermolecular interaction through π-π  interaction.

  1. (Page 11 line 258)“This would be attributed to the lower amount of CAD in CLC3. Due to the high molecular packing in CL3, there would be less CAD incorporation in the CLC3 hybrid. Nevertheless, the interlayer space was expanded by the CAD incorporation.”I think there is a problem with the logic of this statement and there is no good explanation as to why the first figure in Figure 8 shows the opposite phenomenon.

=> We greatly appreciate the reviewer’s point. We found two errors during revision. One is that we made a typo of “CLC1” to “CLC3”, which made a big confusion. The other point is that our explanation was not enough. Basically, we hypothesized the roles of CAD as i) retarding the release by providing π-π interaction and ii) accelerating the release by expanding interlayer space and to accommodate more solvent molecules. For CLC2 and CLC3, the retarding effect seems to dominate accelerating effect. The CLC1 contains only a few amount of CAD as pi-pi mediator, but it happens to expand the interlayer space significantly. In this manner, the accelerating effect of CAD exceeded retarding effect in CLC1, giving rise to more release of CA. We modified the manuscript in order to clearly explain our interpretation as follows.

Original Page 11 line 258) “This would be attributed to the lower amount of CAD in CLC3. Due to the high molecular packing in CL3, there would be less CAD incorporation in the CLC3 hybrid. Nevertheless, the interlayer space was expanded by the CAD incorporation.”

Revised Page 14 line 312) As it was previously reported that the intact layer structure of LDH is well-preserved after payload release [Li. L, et al. pH-Triggered Release and Degradation Mechanism of Layered Double Hydroxides with High Loading Capacity, Advanced Materials Interfaces, 10, (8), 2202396], we could expect that the existence of CAD would not alter the structure of LDH after release but modify the release kinetics of cinnamic acid moiety.

Page 14 line 317) Although the phenomenon is opposite to what we expected from the strong π-π interaction, we could suggest a plausible explanation on this finding based on the access of water molecules to the expanded interlayer space. The amount of CAD is the lowest in CLC1 due to high amount of CA. At the same time, the interlayer distance of CLC1 expanded like in CLC1 or CLC2. The incorporation of CAD has two controversial effects in CA release: i) retarding the release by providing π-π interaction and ii) accelerating the release by expanding interlayer space and to accommodate more solvent molecules. The accelerating effect of CAD in CLC1 would be higher than retarding effect due to the small amount despite of the same lattice expansion compared with CLC2 and CLC3.

  1. (Page 16 line 371)“In a typical run of computations, the simulation box corresponded to 27 unit cells to use an LJ cut-off equal to 1.2 nm.”Why use an LJ cut-off equal to 1.2nm?

=> The use of cut-off equal to 1.2 nm is a general value used for Monte Carlo or Molecular Dynamic simulations. Indeed Lennard-Jones parameters are calculated using interactions between atoms and a large value required a large number of inter-atomic calculations while a too small value for cut-off can fail to reproduce the van der Waals interactions. Another thing is that the value of the cut-off imposes the minimal size of the cell to consider in the calculations, since the length of the cell must be at the minimum twice the value of the cut-off.

  1. Some of the images in this article suggest colorization, the colors are too simple, and some images are not clear enough, Some of the formatting of the article is wrong, as follows:

1)       The curves in Figure 1Figure4Figure5Figure6need to be colored.

2)       The clarity of Figure 7 is too low.

3)       The subheading in the Materials and Methods section is not numbered correctly

=> We appreciate the reviewer’s valuable comments. It is prime importance to represent clear and distinguishable figures. We revised the Fig. 1, 4, 5, 6, and 7 with appropriate coloring and adjustments. We apologize that the resolution of SEM image (Fig. 7) is the maximum at this moment. In order to clarify the particle dimension a bit more, we adjusted brightness and contrast in the revised manuscript.

Furthermore, the subheadings of “Materials and Methods” section were also revised.

Color legend. CA: black; CL1: red; CL2: orange; CL3: yellow; CLC1: cyan; CLC2: blue, CLC3: purple, CAD: green, LDH: gray

Figure 1, 4, 5, 6, and 7 in the revised manuscripts are all in revised version.

Reviewer 2 Report

Comments and Suggestions for Authors

The author did a study on the synthesis of cargo encapsulated LDH nanoparticles. The characterization and simulation were comprehensive enough to prove their structures. I suggest minor revision before publication.

1.       Can the author have a short comparison between LDH and other more widely used nanoparticles in drug delivery? And its pros and cons?

2.       Methodology-wise, can the dose of the encapsulated molecules be tunable?

3.       Line 182: ‘As the CA molecular arrangement was not so effective’ its not a clear statement. Can the author make it more specific?

4.       The wafer amount in interlayer (>150-degree peak) looks like different between the three cases. Any explanation?

5.       in Fig.7, why did the particles form aggregate?  How the particle dimension ~100nm was defined.  It’s very obscure from the image.

6.       In figure8. The cargo release experiment was done in deionized water. Is the learning here easily transferable to any in-vivo situation?

Author Response

Reviewer 2

The author did a study on the synthesis of cargo encapsulated LDH nanoparticles. The characterization and simulation were comprehensive enough to prove their structures. I suggest minor revision before publication.

  1. Can the author have a short comparison between LDH and other more widely used nanoparticles in drug delivery? And its pros and cons?

=> We appreciate the reviewer’s valuable comments. In order to inspire readers, it is important to compare the pros and cons of LDH compared with widely used nanoparticles such as polymers. We added several sentences in the introduction part as follows.

Added Page 2 line 54) Compared with well-known polymer based drug delivery system, LDH has distinguishable advantages and disadvantages. Polymer-based drug delivery materials can be developed in wide variety by modifying the side chains or to arranging monomers. It is easy to control hydrophilicity/hydrophobicity of the material and the release property of polymer is fairly ordered according to the size of particles. However, the degradation of polymer-based delivery system is not controllable, giving rise to unexpected resides and byproduct. On the other hand, LDH-based delivery system does not have flexibility like polymer and it is not simple to prepare them with intended physicochemical properties. Nevertheless, once prepared LDH delivery system can efficiently prevent the payloads from external chemical/physical stimulation; the degradation of LDH results in bioresorbable cations and anions, giving rise to less toxicity. In this regard, the LDH-based delivery system is also attracting interests along with polymer-based systems.

  1. Methodology-wise, can the dose of the encapsulated molecules be tunable?

=> We appreciate the reviewer’s insightful comment. There is a theoretical maximum of encapsulation in LDH as the encapsulation is based on the charge-charge neutralization. It does not mean the dose of encapsulated molecules cannot be tunable. We can adjust the amount of intercalated organic moiety by adding another co-intercalant anions such as nitrate or chlorides. However, it is also still not easy to precisely control the amount of encapsulated molecules through co-intercalation method, as the competitive intercalation between target molecule (cinnamic acid in this study) and controlling molecule (nitrate or chloride) is not fairly controllable. To summarize, it is possible to tune the dose of encapsulated molecules theoretically, but there should be trials and errors to optimize the dose.

  1. Line 182: ‘As the CA molecular arrangement was not so effective’ its not a clear statement. Can the author make it more specific?

=> We agree with the reviewer that the sentence in the original version was not clear. We revised the manuscript as follows.

Original Page 8 line 182) As the CA molecular arrangement was not so effective in LDH gallery space as in CA crystal, the endothermic peak for p-p interaction would be broadened compared with CA only.

Revised Page 9 line 203) As the CA molecules are located in a random manner rather than packing in an ordered array in LDH gallery space as in CA crystal, the endothermic peak for p-p interaction would be broadened compared with CA only.

  1. The water amount in interlayer (>150-degree peak) looks like different between the three cases. Any explanation?

=> We appreciate the reviewer’s advice which we did not realize during data interpretation. Indeed, the DSC diagram (Fig. 5) exhibited that the CL1 has the largest amount of water as the peak area above 150˚C was the biggest for CL1. It is not clearly addressed in this point, why CL1 has the largest interlayer water; however, we could suggest as possible explanation. The cinnamic acid moiety, which was hydrophilic due to the carboxylate moiety, could be easily hydrated. As the CL1 would have the largest amount of cinnamic acid in the interlayer space among the three hybrids, the degree of hydration would also be the highest in CL1. Although the amount of interlayer water is different to each other in CL hybrid, we would like to say that the degree of hydration might not influence the stabilization of CA moiety nor the molecular arrangement in the CL hybrids. In order to address this point, we revised the manuscript as follows.

Added Page 9 line 207) It seems that the CL1 has the largest amount of water as the peak area above 150°C was the biggest for CL1. It is not clearly addressed in this point, why CL1 has the largest interlayer water; however, we could suggest as possible explanation. The cinnamic acid moiety, which was hydrophilic due to the carboxylate moiety, could be easily hydrated. As the CL1 would have the largest amount of cinnamic acid in the interlayer space among the three hybrids, the degree of hydration would also be the highest in CL1. We would like to emphasize, however, that the degree of hydration did not influence the stabilization of CA moiety nor the molecular arrangement in the CL hybrids. highest in CL1. We would like to emphasize, however, that the degree of hydration did not influence the stabilization of CA moiety nor the molecular arrangement in the CL hybrids.

  1. in Fig.7, why did the particles form aggregate?  How the particle dimension ~100nm was defined.  It’s very obscure from the image.

=> The gathering of LDH particles is usually found in coprecipitated LDHs and the degree was more serious in organic intercalated LDH. It is attributed to the strong interparticle interaction through edge-to-edge, face-to-face and edge-to-face interaction. [Gunawan. P et al., Direct control of drug release behavior from layered double hydroxides through particle interactions. Journal of Pharmaceutical sciences 2008, 97, (10), 4367-4378] The coprecipitated LDH particles used to have unsaturated covalent bond at the edge side which terminates either with metal cation or hydroxyl anions. The cation end of one particle could interact with the anion end of another particle to indued edge-to-edge interaction. Similarly, unsaturated hydroxyl anion could have affinity toward LDH layer through edge-to-face manner. Once organic molecules are intercalated, the surface adsorbed moiety of organic part could induce additional affinity through van der Waals interaction. Therefore, the LDH particles in dried state usually agglomerated in a large lumps; however, the agglomeration is reversible, so that the particles could be separated by an appropriate treatment with solvents. We revised the manuscript as follows

Added Page 12 line 257) The agglomeration of LDH is mediated by edge-to-edge, face-to-edge, or face-to-face manner through the unsaturated coordination site and LDH layer [Gunawan.P, Direct control of drug release behavior from layered double hydroxides through particle interactions, Journal of Pharmaceutical Sciences, 97, (10), 4367-6378], and the degree of agglomeration became more significant when organic molecules are intercalated. However, the agglomeration is reversible phenomenon, the particle could be separated upon an appropriate treatment of solvent.

  1. In figure8. The cargo release experiment was done in deionized water. Is the learning here easily transferable to any in-vivo situation?

=> We fully agree with the reviewer that the release test in deionized water only is not enough. For the practical application, release tests in target simulated body fluid are required. In fact, the CA release in this study was originally aimed to the development of cosmetics or food, of which situation is usually based on water with complicated ingredients inside. Different from drug delivery system which targets specific organs like stomach, intestine, or cytoplasm, the real situation of cosmetics or food is fairly difficult to simulate. In this regard, we only carried out release test in the deionized water system. In order to emphasize this point, we modified both “Introduction” and “Results and Discussion” parts as follows:

Original (Intro1) Page 3 line 95) Considering the wide utilization of phenolic acid in drug, nutraceutical or cosmetics, cinnamic acid (CA) was selected as a model organic molecule.

Revised (Intro1) Page 3 line 95) Considering the wide utilization of phenolic acid as biofunctionalized substance, cinnamic acid (CA) was selected as a model organic molecule. Stabilization of CA in an inorganic matrix could be applied to the cosmetics by absorbing ultraviolet rays and to the nutraceuticals by efficiently preserve CA moiety during food process.

Original (Intro2) Page 3 line 104) Then, the time-dependent release profile of CA in deionized water was monitored and the importance of intermolecular interaction between CA and CAD inside the LDH matrix was analyzed.

Revised (Intro2)  Page 3 line 106) Taking into account the potential applications as cosmetics and nutraceutics, the time-dependent release profile of CA in deionized water was monitored and the importance of intermolecular interaction between CA and CAD inside the LDH matrix was analyzed.

Added (Results and Discussion) Page 14 line 328) Based on the release results in neutral pH condition and previous report [Li. L, et al. pH-Triggered Release and Degradation Mechanism of Layered Double Hydroxides with High Loading Capacity, Advanced Materials Interfaces, 10, (8), 2202396], we could expect that the release of CA in acidic pH would be more than 40% within 24 h. As the cinnamic acid stabilized LDH can be utilized in cosmetics by preserving CA moiety during product process, we could assume that the existence of π-π mediator like CAD prevents CA from the unexpected release during manufacturing.

Reviewer 3 Report

Comments and Suggestions for Authors

The manuscript presents an extensive study related to the incorporation of cinnamic acid into Zn/Al LDHs using the coprecipitation method. The authors investigate the consequences of cinnamaldehyde addition and the effects brought by pi-pi interactions between the organic moieties on the expansion of the interlayer region. Overall, the manuscript is well written but it has however some drawbacks that should be solved before publication. 

1. The auhors should emphasize what is the end use of the obtained material (beyond being a model for a theoretical approach)

2. The studies on the release of the cinnamic acid from the LDH matrix were performed in water. However for practical applications, the release of cinnamic acid should be investigated in media similar to those where the hybrid composite would be used (for example if the product is intended to be used in pharmaceutics, the release of cinnamic acid should be determined in aqueous media simulating the gastric fluid or the intestinal fluid, and so on) . Therefore a study at different pH values would be useful.

3. The structural modifications of the solids recovered after the release should be also mentioned.

Comments on the Quality of English Language

In what concerns the English, there are minor editing mistakes that should be corrected

line 36 replace was by were

line 41 replace have by has

line 48 replace has by have

line 101 replace peaks by dffraction lines or reflections - make this modification in all places related to XRD results discussion

lin 105 - "The d-spacing values of CLs were determined 1.85 nm" - Please rephrase 

line 124 interact - add s (interacts)

line 127 Figure 2A ( replace A by a)

line 159 - intact moiety of CA were well - replace were by was or replace moiety by moieties

lines 162-163 It was noteworthy that the ionic property of CA in both CL 162 and CLC hybrids were fairly high to elucidate its strong electrostatic attraction toward ( use either "properties... were" or "property... was")

Line 257: The CA release in CLC1 was more accelerated compared with CL1, which was 257 reverse phenomena with CLC2 and CLC3. - Please rephrase

Author Response

Reviewer 3

The manuscript presents an extensive study related to the incorporation of cinnamic acid into Zn/Al LDHs using the coprecipitation method. The authors investigate the consequences of cinnamaldehyde addition and the effects brought by pi-pi interactions between the organic moieties on the expansion of the interlayer region. Overall, the manuscript is well written but it has however some drawbacks that should be solved before publication. 

  1. The authors should emphasize what is the end use of the obtained material (beyond being a model for a theoretical approach)

=> We appreciate the insightful comment of the reviewer. Although the current work is limited to the theoretical point of molecular arrangement and release, the end-point application should be suggested to catch the interests of the readers as well as to provide importance of the experimental design. In fact, our original design was to utilized CA incorporated LDH for cosmetics or nutraceutics. In both applications, the CA moiety should preserve its immobilized state during production process and release the payloads in a specific condition. Although the current manuscript cannot deal with food processing or cosmetic applications, we should have highlighted the potential utility of the developed hybrids system. In this regard, we revised “Introduction”, “Results and Discussion”, and “Conclusion” as follows.

Original (Intro1) Page 3 line 95) Considering the wide utilization of phenolic acid in drug, nutraceutical or cosmetics, cinnamic acid (CA) was selected as a model organic molecule.

Revised (Intro1) Page 3 line 95) Considering the wide utilization of phenolic acid as biofunctionalized substance, cinnamic acid (CA) was selected as a model organic molecule. Stabilization of CA in an inorganic matrix could be applied to the cosmetics by absorbing ultraviolet rays and to the nutraceuticals by efficiently preserve CA moiety during food process.

Original (Intro2) Page 3 line 104) Then, the time-dependent release profile of CA in deionized water was monitored and the importance of intermolecular interaction between CA and CAD inside the LDH matrix was analyzed.

Revised (Intro2)  Page 3 line 106) Taking into account the potential applications as cosmetics and nutraceutics, the time-dependent release profile of CA in deionized water was monitored and the importance of intermolecular interaction between CA and CAD inside the LDH matrix was analyzed.

Added (Results and Discussion) Page 14 line 328) Based on the release results in neutral pH condition and previous report [Li. L, et al. pH-Triggered Release and Degradation Mechanism of Layered Double Hydroxides with High Loading Capacity, Advanced Materials Interfaces, 10, (8), 2202396], we could expect that the release of CA in acidic pH would be more than 40% within 24 h. As the cinnamic acid stabilized LDH can be utilized in cosmetics by preserving CA moiety during product process, we could assume that the existence of π-π mediator like CAD prevents CA from the unexpected release during manufacturing.

Added (Conclusion) Page 19 line 476) Taking into the potential application of phenolic acids in cosmetics or food ingredients, the utilization of π-π interaction with neutral derivatives could enhance the stability of phenolic acid and provide ease of handling for the production process.

  1. The studies on the release of the cinnamic acid from the LDH matrix were performed in water. However for practical applications, the release of cinnamic acid should be investigated in media similar to those where the hybrid composite would be used (for example if the product is intended to be used in pharmaceutics, the release of cinnamic acid should be determined in aqueous media simulating the gastric fluid or the intestinal fluid, and so on) . Therefore a study at different pH values would be useful.

=> We appreciate the advice from the reviewer. It is correct that the release tests should be carried out in various pH condition to interpret the release behaviors in detail. However, we sincerely apologize to the reviewer that the additional experiments in various pH condition were not carried out as we do not have enough samples to accomplish release tests within a limited time span. Fortunately, the pH dependent release behavior of LDH materials is fairly typical; fast burst followed by slow release in neutral pH and accelerated release in acidic pH [Li. L, et al. pH-Triggered Release and Degradation Mechanism of Layered Double Hydroxides with High Loading Capacity, Advanced Materials Interfaces, 10, (8), 2202396, J. Kim, et al. Journal of Nanoscience and Nanotechnology, 7, (11), 3700-3705]. As the current CL and CLC hybrids are LDH-based organic reservoir, we expect that the release in acidic pH would be more facilitated than in neutral condition. We added this discussion in the Results and Discussion part of release test as follows.

Added (Results and Discussion)  Page 14 line 328) Based on the release results in neutral pH condition and previous report [Li. L, et al. pH-Triggered Release and Degradation Mechanism of Layered Double Hydroxides with High Loading Capacity, Advanced Materials Interfaces, 10, (8), 2202396], we could expect that the release of CA in acidic pH would be more than 40% within 24 h. As the cinnamic acid stabilized LDH can be utilized in cosmetics by preserving CA moiety during product process, we could assume that the existence of π-π mediator like CAD prevents CA from the unexpected release during manufacturing.

  1. The structural modifications of the solids recovered after the release should be also mentioned.

=> We appreciate the comment from the reviewer. The structure of LDH lattice after payload release in neutral pH would be preserved intact, as the release was mediated by the diffusion along the LDH layer. It was also reported that the layer structure of LDH along ab-plane direction was well preserved after intercalant release at neutral and slightly acidic pH [Li. L, et al. pH-Triggered Release and Degradation Mechanism of Layered Double Hydroxides with High Loading Capacity, Advanced Materials Interfaces, 10, (8), 2202396]. Although the layer stacking along c-axis became disorder after the release of intercalated moiety, the general lattice structure would remain unchanged. We also expect that the structure of LDH was preserved after CA release. In this regard, we modified the Result and Discussion part as follows.

Added Page 14 line 312) As it was previously reported that the intact layer structure of LDH is well-preserved after payload release [Li. L, et al. pH-Triggered Release and Degradation Mechanism of Layered Double Hydroxides with High Loading Capacity, Advanced Materials Interfaces, 10, (8), 2202396], we could expect that the existence of CAD would not alter the structure of LDH after release but modify the release kinetics of cinnamic acid moiety.

Thanks to your comment, we also modified English mistakes.